# Expand-and-Randomize: An Algebraic Approach to Secure Computation

**DOI:** 10.3390/e23111461

**Published:** 2021-11-04

**Authors:** Yizhou Zhao, Hua Sun

**Affiliations:** Department of Electrical Engineering, University of North Texas, Denton, TX 76203, USA

**Keywords:** secure computation, capacity, algebraic codes

## Abstract

We consider the secure computation problem in a minimal model, where Alice and Bob each holds an input and wish to securely compute a function of their inputs at Carol without revealing any additional information about the inputs. For this minimal secure computation problem, we propose a novel coding scheme built from two steps. First, the function to be computed is *expanded* such that it can be recovered while additional information might be leaked. Second, a *randomization* step is applied to the expanded function such that the leaked information is protected. We implement this expand-and-randomize coding scheme with two algebraic structures—the finite field and the modulo ring of integers, where the *expansion* step is realized with the *addition* operation and the *randomization* step is realized with the *multiplication* operation over the respective algebraic structures.

## 1. Introduction

Cryptographic primitives are canonical and representative problems that capture the key challenges in understanding the fundamentals of security and privacy, and are essential building blocks for more sophisticated systems and protocols. There is much recent interest in using information theoretic tools to tackle classical cryptographic primitives [1,2,3,4,5,6,7]. Along this line, the focus of this work is on a widely studied primitive in cryptography: secure (multiparty) computation [8].

Secure computation refers to the problem where a number of users wish to securely compute a function on their inputs without revealing any unnecessary information. Interestingly, challenging as it seems, secure computation is always *feasible*, i.e., with at least three users, *any* function can be computed securely in the information theoretic sense [9,10]. We consider the most basic model with honest-but-curious users and no colluding users. This work only considers this basic model, but we note that many other variants have been considered in the literature, which may or may not be feasible depending on the specific assumptions and system parameters of the variant (see, e.g., in [11]). However, what is largely open is how to perform secure computation *optimally*, i.e., efficient secure computation solutions are not known for most cases [6].

The main motivation of this work is to make progress towards constructing efficient secure computation codes. Towards this end, we focus on a minimal model of secure computation, introduced by Feige, Kilian, and Naor in 1994 [12]. In this model (see Figure 1), there are three users: Alice, Bob, and Carol. Alice and Bob have inputs $W_{1}$ and $W_{2}$, respectively, and wish to compute a function $f(W_{1},W_{2})$ at Carol without revealing any additional information about their inputs beyond what is revealed by the function itself. To do so, Alice and Bob share a common random variable *Z* that is independent of the inputs and send codewords $X_{1}$ and $X_{2}$ to Carol, respectively. From $X_{1},X_{2}$, Carol can recover $f(W_{1},W_{2})$ and conditioned on $f(W_{1},W_{2})$, the codewords $X_{1}$ and $X_{2}$ are independent of $W_{1},W_{2}$ so that no additional information is leaked.

The key feature of this formulation is that the communication protocol consists of only one codeword from each party that holds the input (thus non-interactive), while for the general secure computation formulation [9,10], interactive protocols are allowed and typically used. Elemental as it seems, this minimal secure computation problem preserves most challenging features of general secure computation; in particular, feasibility results remain *strong* and optimality results remain *weak*, i.e., any function *f* can be computed securely while the construction of efficient codes remains open in general [12,13]. In this work, we focus exclusively on the original three-party formulation of minimal secure computation [12], but note that many interesting variants have been studied (sometimes under different names to highlight different assumptions) in the literature, e.g., more than three parties [14,15,16], colluding parties [17,18,19,20], other security notions [21], and unresponsive parties [22].

The main contribution of this work is a novel coding scheme that relies on algebraic structures to ensure correctness and security. To illustrate the idea of our coding scheme, let us first consider an example. Suppose Alice and Bob each holds a ternary input, $W_{1},W_{2}\in \left\{0,1,2\right\}$, and wish to compute if $W_{1}$ is equal to $W_{2}$, i.e., $f(W_{1},W_{2})=\mathrm{Yes}$ if $W_{1}=W_{2}$, and $f(W_{1},W_{2})=\mathrm{No}$ otherwise.

As the equality function may not be easily computed in a secure manner, we first *expand* it to a linear function so that it becomes simpler to deal with. As shown in Figure 2, we use the linear function $W_{1}-W_{2}$ over the finite field $F_{3}$ (equivalent to operations modulo 3). For this expansion, we require that the original function can be fully recovered by the expanded function. This is easily verified for this example, where $W_{1}-W_{2}\neq 0$ if and only if $W_{1}$ is not equal to $W_{2}$. This expansion step does not solve the secure computation problem because additional information may be leaked. For example, here Carol should only know if $W_{1}-W_{2}=0$ and is not supposed to learn whether $W_{1}-W_{2}$ is 1 or 2. To prevent this leakage, we invoke another step of *randomization* so that the leaked information by the expanded function becomes confusable and thus protected. For this equality function example, when $W_{1}-W_{2}\neq 0$, we wish to make the result equally likely to be 1 or 2. This is realized by multiplying $W_{1}-W_{2}$ with γ, where γ is uniform over {1,2}. The multiplication operation is also over $F_{3}$. Thus,
(1)$$\begin{array}{c} \mathrm{when}\;W_{1}-W_{2}=1,\gamma \times \left(W_{1}-W_{2}\right)=\gamma \;\mathrm{is}\;\mathrm{equally}\;\mathrm{likely}\;\mathrm{to}\;\mathrm{be}\;1\;\mathrm{or}\;2; \end{array}$$
(2)$$\begin{array}{c} \mathrm{when}\;W_{1}-W_{2}=2,\gamma \times \left(W_{1}-W_{2}\right)=\gamma \times 2\;\mathrm{is}\;\mathrm{equally}\;\mathrm{likely}\;\mathrm{to}\;\mathrm{be}\;1\;\mathrm{or}\;2. \end{array}$$Note that 2×2=4=1 over $F_{3}$. After this randomization step, the *randomized expanded function* does not reveal any additional information beyond the original equality function. The above expand-and-randomize procedure can be easily converted to a distributed secure computation protocol. In particular, Alice and Bob share a common random variable, Z=(γ,z), where γ and *z* are independent, γ is uniform over {1,2}, and *z* is uniform over {0,1,2}. The codewords $X_{1},X_{2}$ sent by Alice and Bob to Carol are
(3)$$\begin{array}{ccc} X_{1} & = & \gamma \times W_{1}+z, \end{array}$$
(4)$$\begin{array}{ccc} X_{2} & = & \gamma \times W_{2}+z. \end{array}$$To decode $f(W_{1},W_{2})$ with no error, Carol subtracts $X_{2}$ from $X_{1}$, $X_{1}-X_{2}=\gamma \times \left(W_{1}-W_{2}\right)$ and claims that $W_{1}$ is equal to $W_{2}$ if and only if $X_{1}-X_{2}=0$. To see why perfect security holds, note that $\left(X_{1},X_{2}\right)$ is invertible to $\left(X_{1}-X_{2},X_{2}\right)$; both $X_{1}-X_{2}$ and $X_{2}$ (protected by an independent uniform noise *z*) do not leak any information. Specifically, the joint distributions of $\left(X_{1},X_{2}\right)$ remain the same for all $\left(W_{1},W_{2}\right)$ pairs so that $f(W_{1},W_{2})$ are the same. That is, when $W_{1}$ is equal to $W_{2}$, i.e., $\left(W_{1},W_{2}\right)\in \left\{\left(0,0\right),\left(1,1\right),\left(2,2\right)\right\}$, $\left(X_{1},X_{2}\right)$ are identically distributed ($X_{1}$ is uniform over {0,1,2} and $X_{2}$ is equal to $X_{1}$) and the same observation holds for all $\left(W_{1},W_{2}\right)$ pairs where $W_{1}$ is not equal to $W_{2}$ ($X_{1}-X_{2}$ is uniform over {1,2}; $X_{2}$ is independent of $X_{1}-X_{2}$ and is uniform over {0,1,2}). Interestingly, this secure computation code is also communication optimal, i.e., the size of $X_{1}$ and $X_{2}$ must be no less than $\log_{2}3$ bits each (required even if there is no security constraint).

A closer inspection of the above scheme reveals that the key is to find an expanded function such that the expanded function outputs corresponding to the same original function output can be randomized to be fully confusable. The first main result of this work is to characterize the structural properties of such *confusable sets* over the finite field $F_{q}$, where *q* is a prime power. The confusable function outputs turn out to be characterized by the property that their discrete logarithms (in exponential representation of the finite field elements) have the same remainder in modular arithmetic. Details will be presented in Section 3.1.

As it turns out, the expand-and-randomize coding scheme is not limited to the finite field. As our second main result, we implement it over the ring of integers modulo *n*, $Z_{n}=\left\{0,1,\cdots ,n-1\right\}$. The ring is equipped with two operations—addition and multiplication—both defined in modulo *n* arithmetic. Let us consider an example to illustrate how $Z_{n}$ is used. Consider the selected-switch function in Figure 3. Alice has a binary input, $W_{1}\in \left\{0,1\right\}$. Bob has a ternary input, $W_{2}\in \left\{0,1,2\right\}$. When $W_{1}\geq W_{2}$, the switch function *f* is OFF and the output is 0 (we may think that the output is not connected to the input, so it is a constant). When $W_{1}<W_{2}$, the switch function *f* is ON and the output is equal to the input vector (all information about $W_{1},W_{2}$ goes through).

Following the expand-and-randomize coding paradigm, we first expand the original function to the addition function over $Z_{6}$ such that it can be fully recovered. Note that to facilitate the construction of the expanded function, here we perform an invertible transformation on the inputs, $W_{1}\to \tilde{W_{1}}\left(0\to 4,1\to 2\right),W_{2}\to \tilde{W_{2}}\left(0\to 0,1\to 2,2\to 5\right)$. The expanded function reveals more information than allowed when the output is 2 or 4. To protect this information, a randomization step is realized by multiplying γ, which is uniform over {1,5}. Now, 2×{1,5}={2,10}={2,4} modulo 6, and 4×{1,5}={4,20}={2,4} modulo 6. Therefore, the expanded function after randomization can be used to produce the following secure computation protocol. The codewords are $X_{1}=\gamma \times \tilde{W_{1}}+z,X_{2}=\gamma \times \tilde{W_{2}}-z$, where Z=(γ,z), γ and *z* are independent, γ is uniform over {1,5}, and *z* is uniform over {0,1,2,3,4,5}. To decode, Carol will compute $X_{1}+X_{2}=\gamma \times \left({\tilde{W}}_{1}+{\tilde{W}}_{2}\right)$. Comparing the original function *f* and the randomized expanded function $\gamma \times (\tilde{W_{1}}+\tilde{W_{2}})$, it is easy to construct the decoding rule based on $X_{1}+X_{2}$ (see Figure 3). Following a straightforward argument as presented above, we may show that the correctness and security constraints are satisfied. Details will be presented in Theorem 1.

From this example, we find that the crux of the scheme is a *partition* of the elements of $Z_{6}$ into several *disjoint* confusable sets such that when any two elements of a confusable set S are multiplied with γ which is uniform over a carefully chosen set (γ is referred to as the *randomizer*), they will produce identically distributed sets of values; specifically, both will produce the confusable set S.
(5)$$\begin{array}{ccc} & & Z_{6}=\left\{0\right\}\cup \left\{1,5\right\}\cup \left\{2,4\right\}\cup \left\{3\right\},\;\gamma \;\mathrm{is}\;\mathrm{uniform}\;\mathrm{over}\;\left\{1,5\right\};\;\mathrm{over}\;Z_{6}:\\ & & 1\times \{1,5\}=5\times \{1,5\}=\{1,5\},2\times \{1,5\}=4\times \{1,5\}=\{2,4\},3\times \{1,5\}=\{3\}. \end{array}$$The main technical challenge is to understand which sets of elements can serve as the randomizer γ and how the ring $Z_{n}$ is partitioned into disjoint confusable sets such that security is guaranteed. For this purpose, we require a few notions from group theory and number theory. Details are presented in Section 3.2. To get a glimpse, consider the above example (see (Equation 5)), where the randomizer γ is from the set of integers that are coprime with 6 (1 and 5 both have no common divisor with 6), and the confusable sets are the sets of integers that have the same greatest common divisor with 6 (e.g., gcd(2,6)=gcd(4,6)=2).

Our proposed coding scheme is inspired by two examples (binary logical AND function and ternary comparison function) presented in Appendix A and Appendix B of the original minimal secure computation paper [12], where modular arithmetic over a prime number *p* is used. Note that for a prime *p*, the algebraic operations (addition and multiplication) in both finite field $F_{p}$ and the ring of integers modulo *p*, $Z_{p}$ are modular arithmetic. Along this line, our work can be viewed as a generalization of the examples from in [12] to a general class of achievable schemes that distill the underlying algebraic structure and work over finite fields and modulo rings of integers with general (non-prime) cardinality.

## 2. Problem Statement

Consider a pair of inputs $\left(W_{1},W_{2}\right)\in \left\{0,1,\cdots ,m_{1}-1\right\}\times \left\{0,1,\cdots ,m_{2}-1\right\}$ and a function $f:\left\{0,1,\cdots ,m_{1}-1\right\}\times \left\{0,1,\cdots ,m_{2}-1\right\}\to \left\{0,1,\cdots ,|f|-1\right\}$. We assume the function *f* is discrete, and use |f| to denote the cardinality of the range of *f*. $W_{1}$ is available to Alice and $W_{2}$ is available to Bob. Alice and Bob also both hold a common random variable *Z* whose distribution does not depend on $W_{1},W_{2}$.

Alice and Bob wish to compute $f(W_{1},W_{2})$ securely. To this end, Alice sends a codeword $X_{1}$ and Bob sends a codeword $X_{2}$ to Carol. $X_{1}$ is a function of $W_{1}$ and *Z*, and has $L_{1}$ bits. $X_{2}$ is a function of $W_{2}$ and *Z*, and has $L_{2}$ bits. The function *f* is known to Alice, Bob, and Carol. As our proposed code will have a fixed length, here we only define fixed-length codes, i.e., $L_{1}$ does not depend on the value of $W_{1}$. In general, variable-length codes might have a lower expected length (see Remark 3).

From $X_{1},X_{2}$, Carol can recover $f(W_{1},W_{2})$ with no error. This is referred to as the correctness constraint. To ensure Carol does not learn anything beyond $f(W_{1},W_{2})$, the following security constraint must be satisfied.
$$\begin{array}{c} \left(\mathrm{Security}\right)\;\;\mathrm{For}\;\mathrm{any}\;\mathrm{joint}\;\mathrm{distribution}\;\mathrm{of}\;\left(W_{1},W_{2}\right),\;I\left(X_{1},X_{2};W_{1},W_{2}|f\left(W_{1},W_{2}\right)\right)=0. \end{array}$$Equivalently, the security constraint can be stated as follows.
(6)$$\begin{array}{c} \mathrm{For}\;\mathrm{any}\;\left(W_{1},W_{2}\right)\;\mathrm{pairs}\;\mathrm{such}\;\mathrm{that}\;f\left(W_{1},W_{2}\right)\;\mathrm{are}\;\mathrm{equal},\\ (X_{1},X_{2})\;\mathrm{are}\;\mathrm{identically}\;\mathrm{distributed}. \end{array}$$

A rate tuple $\left(L_{1},L_{2}\right)$ is said to be achievable if there exists a secure computation scheme, for which the correctness and security constraints are satisfied. The closure of the set of all achievable rate tuples is called the optimal rate region.

The main result of this work is a new achievable scheme for secure computation and the new scheme works for any joint distribution of $\left(W_{1},W_{2}\right)$, so we do not specify explicitly this joint distribution. Further, for simplicity, we introduce the problem statement as a scalar coding problem. Concrete distributions will be given and *L*-length extensions (block inputs) will be considered when they play more significant roles in the results, e.g., when we discuss ϵ-error schemes in Section 4.2 and converse results in Section 3.3.

## 3. The Main Coding Scheme

In this section, we present a novel secure computation code that implements the expand-and-randomize scheme over the finite field $F_{q}$ and the ring of integers modulo *n*, denoted as $Z_{n}$. Let us start with relevant definitions.

**Definition** **1** (Confusable Sets and Randomizer)**.**
*Sets $S_{0},S_{1},S_{2},\cdots$ are called confusable sets if they form a partition of all elements from $F_{q}$ or $Z_{n}$ and there exists a uniform random variable γ over a set $S^{*}\subset F_{q}$ or $Z_{n}$ such that $\forall s\in S_{i}$, γ×s is uniform over $S_{i}$. γ is called the randomizer.*


The requirement in Definition 1 is stronger than what is needed for security. It suffices to have identical (instead of uniform) distributions over some disjoint set (instead of the confusable set). However, for our proposed scheme, it turns out that these relaxations do not lead to improved achievable rate regions such that they are not considered for simplicity. The notions introduced in Definition 1 are closely related to group actions (and orbits) studied in group theory [23]. Along this line, the main effort of this work is to identify a class of group actions that can be used in secure computation and to prove that the class found indeed forms valid group actions. Our proof is relatively elementary, relying on basic number theoretic properties. It is possible to alternatively prove the statements using group actions. We view our main findings as identifying which operations form group actions and their applicability to secure computation (while the proof of validity can be done in various ways). Another related concept that has been studied in cryptography is randomized polynomials [24,25], which also rely on extra randomization to expand the deterministic function to be computed. Our work used different randomization techniques.

**Definition** **2** (Feasible Expanded Function)**.**
*For a function $f(W_{1},W_{2})$, a function $\tilde{f}\left({\tilde{W}}_{1},{\tilde{W}}_{2}\right)={\tilde{W}}_{1}+{\tilde{W}}_{2}$ over $F_{q}$ or $Z_{n}$ is called a feasible expanded function if the mapping between $W_{1}$ and ${\tilde{W}}_{1}$, the mapping between $W_{2}$ and ${\tilde{W}}_{2}$, and the mapping between $f(W_{1},W_{2})$ and the index of the confusable set to which $\tilde{f}\left(\tilde{W_{1}},{\tilde{W}}_{2}\right)$ belongs are all invertible.*


For an example of a feasible expanded function (for the equality function with ternary inputs) over $F_{3}$, see Figure 2. Specifically, $F_{3}=\left\{0,1,2\right\}=S_{0}\cup S_{1}$, where $S_{0}=\left\{0\right\},S_{1}=\left\{1,2\right\}$. γ is uniform over $S^{*}=\left\{1,2\right\}$. γ×1 and γ×2 are both uniformly distributed over $S_{1}$. ${\tilde{W}}_{1}=W_{1}$, ${\tilde{W}}_{2}=-W_{2}$. *f* is the equality function and $\tilde{f}$ is $W_{1}-W_{2}$. The Yes output of *f* is mapped to $S_{0}$ over $\tilde{f}$ and the No output of *f* is mapped to $S_{1}$ over $\tilde{f}$. For an example of a feasible expanded function over $Z_{6}$, see Figure 3.

A feasible expanded function as defined above naturally leads to a correct and secure computation scheme, presented in the following theorem.

**Theorem** **1.**
*For any function $f(W_{1},W_{2})$, if we have a feasible expanded function $\tilde{f}\left({\tilde{W}}_{1},{\tilde{W}}_{2}\right)={\tilde{W}}_{1}+{\tilde{W}}_{2}$ over $F_{q}$ or $Z_{n}$, then the following computation code is both correct and secure:*

(7)
$$\begin{array}{c} X_{1}=\gamma \times {\tilde{W}}_{1}+z,X_{2}=\gamma \times {\tilde{W}}_{2}-z \end{array}$$

*where Z=(z,γ), γ is the randomizer, z and γ are independent, and z is uniform over $F_{q}$ or $Z_{n}$. Specifically, in this scheme, Alice and Bob each sends a symbol from $F_{q}$ or $Z_{n}$ to Carol.*


**Proof** **of**  **Theorem** **1.**The proof of correctness and security follows in a straightforward manner from the definitions of the confusable sets, the randomizer, and the feasible expanded function. First, we consider the correctness constraint. To recover $f(W_{1},W_{2})$ with no error, Carol may compute $X_{1}+X_{2}=\gamma \times \left({\tilde{W}}_{1}+{\tilde{W}}_{2}\right)$, from which Carol can uniquely identify the index of the confusable set (invertible to the original function output). Note that by the definition of the confusable sets and the randomizer, multiplying with γ does not change the confusable set index. Second, we consider the security constraint (Equation 6). Consider any $\left(W_{1},W_{2}\right)$ pairs that produce the same $f(W_{1},W_{2})$ output, and we show that $\left(X_{1},X_{2}\right)$ are identically distributed. To see this, note that $\left(X_{1},X_{2}\right)$ is invertible to $\left(X_{1}+X_{2},X_{2}\right)=\left(\gamma \times \left({\tilde{W}}_{1}+{\tilde{W}}_{2}\right),\gamma \times {\tilde{W}}_{2}-z\right)$. By the definition of the confusable sets, $X_{1}+X_{2}$ is uniform over the confusable set that corresponds to $f(W_{1},W_{2})$; $X_{2}$ is independent of $X_{1}+X_{2}$ and is uniform over $F_{q}$ or $Z_{n}$ due to the uniformity and independence of *z*. Therefore, $\left(X_{1}+X_{2},X_{2}\right)$ are always uniform thus are identically distributed (so are $\left(X_{1},X_{2}\right)$). The proof is complete. □

The coding scheme in Theorem 1 relies on the structure of the confusable sets and the randomizer upon which feasible expanded functions are built. Thus, it is crucial to understand the structure of the confusable sets and the randomizer, i.e., which set of elements can be used as the randomizer and how the algebraic object is partitioned to confusable sets. This structure problem is addressed next, through algebraic characterizations. The finite field case is considered in Section 3.1 and the ring of integers modulo *n* case is considered in Section 3.2.

### 3.1. Finite Field

We first recall some basic facts of finite fields (refer to standard textbooks such as that in [26]). A finite field $F_{q}$ exists only when $q=p^{n}$, where *p* is a prime and *n* is a positive integer. $F_{q}$ has $q=p^{n}$ elements. Any two fields with $p^{n}$ elements are isomorphic, thus $F_{q}$ is referred to as *the* finite field. The $p^{n}$ elements of $F_{q}$ are the polynomials $a_{0}+a_{1}x+a_{2}x^{2}+\cdots a_{n-1}x^{n-1}$, where $a_{i}\in \left\{0,1,\cdots ,p-1\right\},\forall i\in \left\{0,1,\cdots ,n-1\right\}$. The addition and multiplication operations over $F_{q}$ are defined modulo h(x), where h(x) is an irreducible polynomial of degree *n* that always exists. The non-zero elements of $F_{q}$ form a multiplicative group, denoted as $F_{q}^{\times }$. $F_{q}^{\times }$ is a cyclic group $\left\{1,g,g^{2},\cdots ,g^{q-2}\right\}$ that can be generated by a primitive element $g\in F_{q}^{\times }$. Denote $g^{0}=1$.

**Example** **1.**
*The finite field $F_{2^{3}}$ can be constructed by addition and multiplication modulo $h\left(x\right)=x^{3}+x+1$. The multiplicative group $F_{2^{3}}^{\times }=\left\{1,x,x+1,x^{2},x^{2}+1,x^{2}+x,x^{2}+x+1\right\}$ can be generated by g=x.*

(8)
$$\begin{array}{ccc} & & g^{2}=x^{2},g^{3}=x^{3}\;\mathrm{mod}\;\left(x^{3}+x+1\right)=x+1,g^{4}=x^{2}+x, \end{array}$$


(9)
$$\begin{array}{cc} & g^{5}=x^{3}+x^{2}\;\mathrm{mod}\;\left(x^{3}+x+1\right)=x^{2}+x+1, \end{array}$$


(10)
$$\begin{array}{cc} & g^{6}=x^{3}+x^{2}+x\;\mathrm{mod}\;\left(x^{3}+x+1\right)=x^{2}+1, \end{array}$$


(11)
$$\begin{array}{c} g^{7}=x^{3}+x\;\mathrm{mod}\;\left(x^{3}+x+1\right)=1=g^{0}. \end{array}$$



Equipped with the above results (in particular, the cyclic property of the multiplicative group $F_{q}^{\times }$), we are ready to the state in the following theorem the algebraic characterization of the confusable sets and the randomizer over $F_{q}$.

**Theorem** **2.**
*For $F_{q}$ where $q=p^{n}$, p is a prime, n is an integer, and g is a primitive element of $F_{q}^{\times }$, the confusable sets and the randomizer can be chosen as follows. Consider any divisor d of $p^{n}-1$, i.e., $b=(p^{n}-1)/d$ is an integer.*

(12)
$$\begin{array}{c} \gamma \;\mathrm{is}\;\mathrm{uniform}\;\mathrm{over}\;S^{*}=\left\{g^{0},g^{d},g^{2d},\cdots ,g^{(b-1)d}\right\}, \end{array}$$


(13)
$$\begin{array}{c} F_{q}=S_{0}\cup S_{1}\cup S_{2}\cup \cdots \cup S_{d}, \end{array}$$


(14)
$$\begin{array}{c} S_{0}=\left\{0\right\},S_{i}=\left\{g^{i-1},g^{d+i-1},g^{2d+i-1},\cdots ,g^{(b-1)d+i-1}\right\},\forall i\in \left\{1,2,\cdots ,d\right\}. \end{array}$$



**Remark** **1.**
*In words, the elements of a confusable set are such that their discrete logarithms have the same remainder modulo a divisor of $p^{n}-1$.*


Before we prove Theorem 2, let us first understand it through an example and use it to securely compute a function.

**Example** **2.**
*Consider $F_{7}=\left\{0,1,2,3,4,5,6\right\}$. A primitive element of $F_{7}^{\times }$ is 3. Setting d=3, the following confusable sets are given by Theorem 2.*

(15)
$$\begin{array}{c} S_{0}=\left\{0\right\},S_{1}=\left\{3^{0},3^{3}\right\}=\left\{1,27\right\}\;\mathrm{mod}\;7=\left\{1,6\right\},\\ S_{2}=\left\{3^{1},3^{4}\right\}=\left\{3,4\right\},S_{3}=\left\{3^{2},3^{5}\right\}=\left\{2,5\right\}. \end{array}$$

*Consider the function $f(W_{1},W_{2})$ shown in Figure 4, for which a feasible expanded function can be built upon the confusable sets given above.*


While the primitive element *g* of $F_{q}^{\times }$ is guaranteed to exist, there is no analytic formula for it and finding it computationally is difficult in general. Further, given the polynomial representation of *g*, it is generally non-trivial to determine the minimum field size *q* such that there exists a feasible expanded function over $F_{q}$ for a specific function *f*. A list of confusable sets for all finite fields $F_{q},q<20$ is given in Figure A1 (see the Appendix A).

**Proof** **of** **Theorem** **2.**To verify that the definition of the confusable sets is satisfied, we only need to show that $\forall s\in S_{i},i\in \left\{0,1,\cdots ,d\right\}$, γ×s is uniform over $S_{i}$. This is proved as follows. When i=0, $S_{0}=\left\{0\right\}$ so that s=0 and $\gamma \times s=\left\{0\right\}=S_{0}$. When i∈{1,⋯,d}, consider any element from $S_{i}$, e.g., $s=g^{jd+i-1},j\in \left\{0,1,\cdots ,b-1\right\}$. We have
(16)$$\begin{array}{ccc} \gamma \times s & = & \left\{g^{0},g^{d},g^{2d},\cdots ,g^{(b-1)d}\right\}\times g^{jd+i-1} \end{array}$$
(17)$$\begin{array}{ccc} & = & \left\{g^{jd+i-1},g^{(j+1)d+i-1},g^{(j+2)d+i-1},\cdots ,g^{(j+b-1)d+i-1}\right\} \end{array}$$
(18)$$\begin{array}{ccc} & = & \left\{g^{i-1},g^{d+i-1},g^{2d+i-1},\cdots ,g^{(b-1)d+i-1}\right\}=S_{i} \end{array}$$
where (Equation 18) follows from the fact that $g^{bd}=g^{p^{n}-1}=1$ and the observation that any *b* consecutive integers form the same set under modulo *b*, i.e., {0,1,⋯,b−1}={j,j+1,⋯,j+b−1}modb. As γ is uniform, γ×s is uniform (over $S_{i}$) as well. □

### 3.2. Ring of Integers Modulo n

To facilitate the presentation of the algebraic characterization of the confusable sets and the randomizer over $Z_{n}$, we first introduce some definitions and preliminary results.

**Definition** **3** (Set of Integers with Same gcd)**.**
*Consider any proper divisor d of a given integer n, i.e., d<n and n/d is an integer. We denote by $Z_{n}^{\left(d\right)}$ the set of integers in $Z_{n}$ so that their greatest common divisors with n are d, i.e., $Z_{n}^{\left(d\right)}=\left\{a\in Z_{n}|\gcd\left(a,n\right)=d\right\}$.*


For example, suppose n=15=3×5, which has proper divisors 1, 3, 5. Then,
(19)$$\begin{array}{ccc} & & Z_{15}^{\left(1\right)}=\left\{1,2,4,7,8,11,13,14\right\},Z_{15}^{\left(3\right)}=\left\{3,6,9,12\right\},Z_{15}^{\left(5\right)}=\left\{5,10\right\}. \end{array}$$
(20)$$\begin{array}{ccc} & & \mathrm{Further},\;Z_{15}=\left\{0,1,\cdots ,14\right\}=\left\{0\right\}\cup Z_{15}^{\left(1\right)}\cup Z_{15}^{\left(3\right)}\cup Z_{15}^{\left(5\right)}. \end{array}$$The set $Z_{n}^{\left(1\right)}$ has been extensively studied in abstract algebra (see e.g., [23]) and number theory (see, e.g., in [27]), and is referred to as the multiplicative group of integers modulo *n* (it turns out to form a group under multiplication modulo *n*), so we adopt the standard existing notation $Z_{n}^{\times }=Z_{n}^{\left(1\right)}$.

Note that
(21)$$\begin{array}{c} Z_{n}^{\left(d\right)}=\left\{a\in Z_{n}|\gcd\left(a,n\right)=d\right\}=d\times \left\{a\in Z_{n/d}|\gcd\left(a,n/d\right)=1\right\}=d\times Z_{n/d}^{\times }. \end{array}$$For example,
(22)$$\begin{array}{c} Z_{15}^{\left(3\right)}=\left\{3,6,9,12\right\}=3\times \left\{1,2,3,4\right\}=3\times Z_{5}^{\times },\\ Z_{15}^{\left(5\right)}=\left\{5,10\right\}=5\times \left\{1,2\right\}=5\times Z_{3}^{\times }. \end{array}$$

We present an important result on the projection of a multiplicative subgroup of $Z_{n}^{\times }$ over $Z_{d}^{\times }$ in the following lemma. To differentiate set and multiset (where an element might appear several times), we use the notation $\bar{\{}H\bar{\}}$ for a multiset *H*.

**Lemma** **1.**
*Consider an arbitrary subgroup $G_{n}$ of $Z_{n}^{\times }$ (under multiplication modulo n). When we take $G_{n}$ modulo d (where d is a divisor of n and d≠1), we have multiple copies of a subgroup of $Z_{d}^{\times }$ (under multiplication modulo d), i.e., $G_{n}\;\mathrm{mod}\;d=\bar{\{}G_{d},G_{d},\cdots ,G_{d}\bar{\}}$, where $G_{d}$ is a subgroup of $Z_{d}^{\times }$.*


In Lemma 1, *G* denotes a subgroup and the subscript specifies the original group. The proof of Lemma 1 is presented in Section 3.4. Here, for illustration, we give an example.

**Example** **3.**
*Consider a subgroup $G_{15}=\left\{1,11\right\}$ of $Z_{15}^{\times }$. We have $G_{15}\;\mathrm{mod}\;3=\left\{1,2\right\}$ so that $G_{3}=\left\{1,2\right\}$, which is a subgroup of (in fact, equal to) $Z_{3}^{\times }=\left\{1,2\right\}$. $G_{15}\;\mathrm{mod}\;5=\bar{\{}1,1\bar{\}}$, which is two copies of $G_{5}=\left\{1\right\}$, and $G_{5}$ is a (trivial) subgroup of $Z_{5}^{\times }=\left\{1,2,3,4\right\}$.*

*Consider another subgroup $G_{15}=\left\{1,4,11,14\right\}$ of $Z_{15}^{\times }$. $G_{15}\;\mathrm{mod}\;3=\bar{\{}1,1,2,2\bar{\}}$, which is two copies of $G_{3}=\left\{1,2\right\}=Z_{3}^{\times }$. $G_{15}\;\mathrm{mod}\;5=\bar{\{}1,4,1,4\bar{\}}$, which is two copies of $G_{5}=\left\{1,4\right\}$ and $G_{5}$ is a subgroup of $Z_{5}^{\times }=\left\{1,2,3,4\right\}$.*

*Consider $G_{15}=Z_{15}^{\times }=\left\{1,2,4,7,8,11,13,14\right\}$. $G_{15}\;\mathrm{mod}\;3=\bar{\{}1,2,1,1,2,2,1,2\bar{\}}$, which is 4 copies of $G_{3}=\left\{1,2\right\}=Z_{3}^{\times }$. $G_{15}\;\mathrm{mod}\;5=\bar{\{}1,2,4,2,3,1,3,4\bar{\}}$, which is 2 copies of $G_{5}=\left\{1,2,3,4\right\}=Z_{5}^{\times }$.*


Given a subgroup $G_{d}$ of the group $Z_{d}^{\times }$, we may partition $Z_{d}^{\times }$ into cosets (see, e.g., Proposition 4 in Chapter 3 of [23] or Theorem 6.2 of [28]). Setting *d* as n/d, we have that $Z_{n/d}^{\times }$ may be partitioned into cosets with $G_{n/d}$. Combining with (Equation 21), i.e., $Z_{n}^{\left(d\right)}=d\times Z_{n/d}^{\times }$, we may partition $Z_{n}^{\left(d\right)}$ into cosets with $G_{n/d}$. This partition is denoted by $Z_{n}^{\left(d\right)}/G_{n/d}$.

**Example** **4.**
*Continuing from Example 3, consider a subgroup $G_{15}=\left\{1,11\right\}$ of $Z_{15}^{\times }$. Then,*

(23)
$$\begin{array}{c} Z_{15}^{\times }/G_{15}=\left\{1,11\right\}\cup \left\{2,7\right\}\cup \left\{4,14\right\}\cup \left\{8,13\right\} \end{array}$$

*where the partition is obtained from the cosets, e.g., {2,7}=2×{1,11}=7×{1,11} is a coset of $G_{15}$ with representative $2\in Z_{15}^{\times }$ or $7\in Z_{15}^{\times }$. Similarly, when $G_{15}=\left\{1,11\right\}$, from Example 3 we have $G_{3}=\left\{1,2\right\},G_{5}=\left\{1\right\}$ and the partitions $Z_{15}^{\left(5\right)}/G_{3},Z_{15}^{\left(3\right)}/G_{5}$ are as follows:*

(24)
$$\begin{array}{ccc} & & Z_{15}^{\left(5\right)}/G_{3}=5\times \left\{1,2\right\},\\ & & Z_{15}^{\left(3\right)}/G_{5}=3\times \left\{\{1\}\cup \{2\}\cup \{3\}\cup \{4\}\right\}. \end{array}$$


*For another choice of $G_{15}$ (again from Example 3), consider $G_{15}=\left\{1,4,11,14\right\}$ of $Z_{15}^{\times }$. Then, from Example 3, $G_{3}=\left\{1,2\right\},G_{5}=\left\{1,4\right\}$. The partitions are*

(25)
$$\begin{array}{ccc} & & d=1:Z_{15}^{\times }/G_{15}=\left\{1,4,11,14\right\}\cup \left\{2,7,8,13\right\},\\ & & d=3:Z_{15}^{\left(3\right)}/G_{5}=3\times \left\{\{1,4\}\cup \{2,3\}\right\},\\ & & d=5:Z_{15}^{\left(5\right)}/G_{3}=5\times \left\{1,2\right\}. \end{array}$$


*For the final choice of $G_{15}=Z_{15}^{\times }$ from Example 3, we have $G_{3}=Z_{3}^{\times },G_{5}=Z_{5}^{\times }$ and the partitions are trivial- $Z_{15}^{\times }/G_{15}=Z_{15}^{\times }$, $Z_{15}^{\left(3\right)}/G_{5}=3\times Z_{5}^{\times }$, and $Z_{15}^{\left(5\right)}/G_{3}=5\times Z_{3}^{\times }$.*


The collection of the cosets $Z_{n}^{\left(d\right)}/G_{n/d}$ for all proper divisors *d* is a feasible choice of the confusable sets. This result is stated in the following theorem.

**Theorem** **3.**
*For $Z_{n}$, the confusable sets and the randomizer can be chosen as follows. Consider the set of all proper divisors of n, $\left\{d_{1}=1,d_{2},\cdots ,d_{b}\right\}$ and an arbitrary subgroup $G_{n}$ of $Z_{n}^{\times }$.*

(26)
$$\begin{array}{ccc} & & \gamma \;\mathrm{is}\;\mathrm{uniform}\;\mathrm{over}\;S^{*}=G_{n}, \end{array}$$


(27)
$$\begin{array}{ccc} & & Z_{n}=S_{0}\cup S_{1}\cup S_{2}\cup \cdots =\left\{0\right\}\cup Z_{n}^{\times }/G_{n}\cup Z_{n}^{\left(d_{2}\right)}/G_{n/d_{2}}\cup \cdots \cup Z_{n}^{\left(d_{b}\right)}/G_{n/d_{b}}. \end{array}$$



Before presenting the proof of Theorem 3, we first give an example to illustrate its meaning.

**Example** **5.**
*Continuing from Example 4, consider a subgroup $G_{15}=\left\{1,11\right\}$ of $Z_{15}^{\times }$. Then, from Theorem 3, the confusable sets are*

(28)
$$\begin{array}{ccc} Z_{15} & = & \left\{0\right\}\cup Z_{15}^{\times }/G_{15}\cup Z_{15}^{\left(3\right)}/G_{5}\cup Z_{15}^{\left(5\right)}/G_{3} \end{array}$$


(29)
$$\begin{array}{ccc} & \overset{\left(23)(24\right)}{=} & \left\{0\}\cup \{1,11\}\cup \{2,7\}\cup \{4,14\}\cup \{8,13\}\cup \{3\}\cup \{6\}\cup \{9\right\}\\ & & \;\cup \{12\}\cup \{5,10\}. \end{array}$$

*For each of the confusable set above, it is easy to verify that when an element is multiplied with γ (uniform over $G_{15}$), the result is uniform over the confusable set.*

(30)
$$\begin{array}{c} \left\{2,7\right\}=\gamma \times 2=\gamma \times 7,\;\left\{5,10\right\}=\gamma \times 5=\gamma \times 10,\;\gamma \times 3=\left\{1,11\right\}\times 3=\bar{\{}3,3\bar{\}}. \end{array}$$

*For another example, consider $G_{15}=\left\{1,4,11,14\right\}$. The confusable sets are*

(31)
$$\begin{array}{ccc} Z_{15} & = & \left\{0\right\}\cup Z_{15}^{\times }/G_{15}\cup Z_{15}^{\left(3\right)}/G_{5}\cup Z_{15}^{\left(5\right)}/G_{3} \end{array}$$


(32)
$$\begin{array}{ccc} & \overset{\left(25\right)}{=} & \{0\}\cup \{1,4,11,14\}\cup \{2,7,8,13\}\cup \{3,12\}\cup \{6,9\}\cup \{5,10\}. \end{array}$$

*Let us also verify that the uniform property holds. γ is over $G_{15}=\left\{1,4,11,14\right\}$. For example, consider 7∈{2,7,8,13}, then we have γ×7={7,28,77,98}mod15={7,13,2,8}. Consider 12∈{3,12}, then we have $\gamma \times 12=\bar{\{}12,48,132,168\bar{\}}\;\mathrm{mod}\;15=\bar{\{}12,3,12,3\bar{\}}$, which is 2 copies of {3,12}.*

*Finally, consider $G_{15}=Z_{15}^{\times }$. The confusable sets are $Z_{15}=\left\{0\right\}\cup Z_{15}^{\times }\cup Z_{15}^{\left(3\right)}\cup Z_{15}^{\left(5\right)}$. For any element in $Z_{15}^{\left(3\right)}$, say 6, we have $\gamma \times 6=Z_{15}^{\times }\times 6$$=\bar{\{}6,12,9,12,3,6,3,9\bar{\}}$.*


**Proof** **of** **Theorem** **3.**The proof relies on Lemma 1 and the property of cosets. First, the confusable sets form a partition of $Z_{n}$. Second, we verify the uniform property, i.e., $\forall s\in S_{i}$, γ×s is uniform over $S_{i}$. Consider any $S_{i}$, e.g., a set from $Z_{n}^{\left(d_{i}\right)}/G_{n/d_{i}},i\in \left\{1,\cdots ,b\right\}$. From the construction of $S_{i}$, we have $S_{i}\times 1/d_{i}$ is a coset of $G_{n/d_{i}}$ in $Z_{n/d_{i}}^{\times }$. By the definition of cosets and the fact that $s/d_{i}\in S_{i}\times 1/d_{i}$, we have
(33)$$\begin{array}{c} S_{i}\times 1/d_{i}=G_{n/d_{i}}\times s/d_{i}. \end{array}$$Next, consider
(34)$$\begin{array}{ccc} \left(G_{n}\times s/d_{i}\right)\;\mathrm{mod}\;n/d_{i} & = & \left(\left(G_{n}\;\mathrm{mod}\;n/d_{i}\right)\times s/d_{i}\right)\;\mathrm{mod}\;n/d_{i} \end{array}$$
(35)$$\begin{array}{ccc} & \overset{\mathrm{Lemma}\;1}{=} & \left(\bar{\{}G_{n/d_{i}},\cdots ,G_{n/d_{i}}\bar{\}}\times s/d_{i}\right)\;\mathrm{mod}\;n/d_{i} \end{array}$$
(36)$$\begin{array}{ccc} & \overset{\left(33\right)}{=} & \bar{\{}S_{i}\times 1/d_{i},\cdots ,S_{i}\times 1/d_{i}\bar{\}}\;\mathrm{mod}\;n/d_{i} \end{array}$$
(37)$$\begin{array}{ccc} \Rightarrow \;\;\gamma \times s=G_{n}\times s & = & d_{i}\times \left(G_{n}\times s/d_{i}\right) \end{array}$$
(38)$$\begin{array}{ccc} & \overset{\left(36\right)}{=} & \bar{\{}S_{i},\cdots ,S_{i}\bar{\}}. \end{array}$$Therefore, γ×s is uniform over $S_{i}$. The proof is complete. □

**Remark** **2.**
*From Theorem 3, we see that any subgroup of $Z_{n}^{\times }$ can induce a feasible choice of the confusable sets and the randomizer. We list all possible confusable sets for $Z_{n},n<20$ in Figure A2 (see the Appendix B). We also include in the Appendix C some discussion on the structures of the subgroups of $Z_{n}^{\times }$, based on existing group theory and number theory results.*


### 3.3. Converse

One of the challenges to understand the optimality of secure computation codes is the lack of converse results. In information theory, converse results are statements of impossibility claims and are used to prove optimality. As a starting point, we compare our achievable scheme with existing converse results with no security constraint (i.e., the pure computation problem). Interestingly, when the size of the underlying field or ring is the same as the input size, the scheme in Theorem 1 achieves the information theoretically optimal rate region. Without loss of generality, for secure computation problems, we assume there are no identical rows or columns in the function table (as Carol cannot learn anything about the exact row or column index of such identical rows and columns).

**Proposition** **1.**
*Consider independent and uniform inputs, i.e., $W_{1},W_{2}$ are independent and uniform over {0,1,⋯,m−1}. For a function $f(W_{1},W_{2})$, if a feasible expanded function exists over $F_{q}$ or $Z_{n}$ where q=m or n=m, then the scheme in Theorem 1 is information theoretically optimal.*


Achievability directly follows from Theorem 1 and converse $\left(H\left(X_{1}\right),H\left(X_{2}\right)\geq \log_{2}m\right)$ follows from a simple observation that when there is no security constraint, Alice (Bob) needs to tell Carol the exact value of $W_{1}$ ($W_{2}$). The reason is that otherwise two $W_{1}\left(W_{2}\right)$ will be mapped to the same codeword $X_{1}\left(X_{2}\right)$ and $f(W_{1},W_{2})$ has no identical rows or columns such that some value of $f(W_{1},W_{2})$ cannot be decoded correctly. This (and more general) result has been proved in several different contexts in the literature, see, e.g., the classical function computation of correlated sources work by Han and Kobayashi [29] (Lemma 1) and the recent generalization [30], the computation over multiple access channel work [31] (Lemma 1), and the network coding for computing work [32,33]. Note that the converse holds for block inputs as well, where the rate is defined as the number of bits in the codeword per input symbol. As eliminating the security constraint cannot help, the same converse holds for the secure computation problem as well.

Note that Proposition 1 characterizes the optimal rate region for a class of secure computation problems (which contain infinite instances). One could start from the confusable sets of $F_{q}$ or $Z_{n}$ and invert them into a function $f(W_{1},W_{2})$ with input size m=q or *n*. Functions constructed from this method satisfy Proposition 1 and thus we obtain the optimal rate region.

To the best of our knowledge, the only existing information theoretic converse results for the secure computation problem are the ones obtained in [6], whose expression involves common information terms and an optimization over a class of distributions so that the exact bound needs to be evaluated for each individual instance and is generally not trivial to compute. Interestingly, for some small instances, we find that our achievable scheme is information theoretically optimal (see Remark 3 of Example 6 and Remark 4 of Example 7). For most cases, however, there is a gap in the rate region between the achievable scheme in Theorem 1 and the converse results from in [6], while it is not clear if and by how much the scheme and the converse can be improved. The model considered in [6] is the general secure computation problem that allows interactive multi-round protocols. Therefore, the converse results therein might be generally too strong for the minimal secure computation problem. We note that there are instances where we know better schemes than that in Theorem 1 (see Examples 9 and 10 in the discussion section).

### 3.4. Proof of Lemma 1

The proof of Lemma 1 consists of two parts.

First, we show that the set of elements of $G_{n}\;\mathrm{mod}\;d$, $G_{d}$, forms a subgroup of $Z_{d}^{\times }$. This is proved by two claims: (1) $G_{d}\subset Z_{d}^{\times }$ and (2) $G_{d}$ is closed under multiplication modulo *d*. Note that for finite groups, the verification of subgroups only requires the check of the closure property (i.e., associativity and the existence of identity and inverse elements are automatically guaranteed. Refer to Proposition 1 in Chapter 2 of [23]).

For (1), note that any element *g* of $G_{n}$ belongs to $Z_{n}^{\times }$, so gcd(g,n)=1. As *d* is a divisor of *n*, we have gcd(g,d)=1 and gcd(gmodd,d)=1. Thus, gmodd of $G_{d}$ belongs to $Z_{d}^{\times }$ and $G_{d}\subset Z_{d}^{\times }$.

For (2), consider any two elements of $G_{d}$, e.g., $g_{1},g_{2}\in G_{n}$ and $g_{1}\;\mathrm{mod}\;d,g_{2}\;\mathrm{mod}\;d\in G_{d}$. As $G_{n}$ forms a group, we have for some $g_{3}\in G_{n}$, $\left(g_{1}\times g_{2}\right)\;\mathrm{mod}\;n=g_{3}$, i.e., $g_{1}\times g_{2}=k\times n+g_{3}$ for some integer *k*. Then,
(39)$$\begin{array}{ccc} \left(\left(g_{1}\;\mathrm{mod}\;d\right)\times \left(g_{2}\;\mathrm{mod}\;d\right)\right)\;\mathrm{mod}\;d & = & (g_{1}\times g_{2})\;\mathrm{mod}\;d \end{array}$$
(40)$$\begin{array}{ccc} & = & (k\times n+g_{3})\;\mathrm{mod}\;d \end{array}$$
(41)$$\begin{array}{ccc} & = & g_{3}\;\mathrm{mod}\;d\;\;\;\;\left(d\;\mathrm{is}\;a\;\mathrm{divisor}\;\mathrm{of}\;n\right) \end{array}$$
(42)$$\begin{array}{ccc} & \in & G_{d} \end{array}$$Therefore $G_{d}$ is closed under multiplication.

Second, we show that in the multiset $G_{n}\;\mathrm{mod}\;d$, each element of $G_{d}$ appears for the same number of times. Denote $G_{n}=\left\{g_{1},g_{2},\cdots ,g_{T}\right\}$. As $G_{n}$ is a subgroup of $Z_{n}^{\times }$, we have
(43)$$\begin{array}{c} \forall i\in \left\{1,\cdots ,T\right\},\left(G_{n}\times g_{i}\right)\;\mathrm{mod}\;n=\left\{g_{1}\times g_{i},g_{2}\times g_{i},\cdots ,g_{T}\times g_{i}\right\}\;\mathrm{mod}\;n=G_{n}. \end{array}$$Denote the multiset ${\bar{G}}_{d}=G_{n}\;\mathrm{mod}\;d=\bar{\{}h_{1},\cdots ,h_{1},h_{2},\cdots ,h_{Q}\bar{\}}$, where $h_{q},q\in \left\{1,\cdots ,Q\right\}$ appears $|h_{q}|$ times and $\forall q_{1}\neq q_{2},h_{q_{1}}\neq h_{q_{2}}$. Assume without loss of generality that $|h_{1}\left|\geq \right|h_{2}\left|\geq \cdots \right|h_{Q}|$. We need to show that $|h_{1}\left|=\right|h_{Q}|$. This proof is presented next.

From the first part of the proof, we know that $G_{d}=\left\{h_{1},h_{2},\cdots ,h_{Q}\right\}$ is a subgroup of $Z_{d}^{\times }$. Applying (Equation 43) to $G_{d}$ and $Z_{d}^{\times }$, we have
(44)$$\begin{array}{c} \forall j\in \left\{1,\cdots ,Q\right\},\left(G_{d}\times h_{j}\right)\;\mathrm{mod}\;d=\left\{h_{1}\times h_{j},h_{2}\times h_{j},\cdots ,h_{Q}\times h_{j}\right\}\;\mathrm{mod}\;d=G_{d}. \end{array}$$Further, setting j=1 in (Equation 44), we have
(45)$$\begin{array}{c} \left(G_{d}\times h_{1}\right)\;\mathrm{mod}\;d=\left\{h_{1}\times h_{1},h_{2}\times h_{1},\cdots ,h_{Q}\times h_{1}\right\}\;\mathrm{mod}\;d=G_{d}=\left\{h_{1},\cdots ,h_{Q}\right\}. \end{array}$$Note that multiplication mod *d* is commutative. Then, there exists $j^{*}\in \left\{1,\cdots ,Q\right\}$ such that
(46)$$\begin{array}{c} \left(h_{j^{*}}\times h_{1}\right)\;\mathrm{mod}\;d=\left(h_{1}\times h_{j^{*}}\right)\;\mathrm{mod}\;d=h_{Q}. \end{array}$$As $h_{j^{*}}\in G_{d}$, there exists $i^{*}\in \left\{1,\cdots ,T\right\}$ such that
(47)$$\begin{array}{c} g_{i^{*}}\;\mathrm{mod}\;d=h_{j^{*}}. \end{array}$$On the one hand,
(48)$$\begin{array}{ccc} ({\bar{G}}_{d}\times h_{j^{*}})\;\mathrm{mod}\;d & = & \bar{\{}\underset{|h_{1}|\mathrm{times}}{\underset{︸}{h_{1}\times h_{j^{*}},\cdots ,h_{1}\times h_{j^{*}}}},h_{2}\times h_{j^{*}},\cdots ,h_{Q}\times h_{j^{*}}\bar{\}}\;\mathrm{mod}\;d\\ & \overset{\left(46)(44\right)}{=} & \bar{\{}\underset{|h_{1}|\mathrm{times}}{\underset{︸}{h_{Q},\cdots ,h_{Q}}},h_{1},\cdots ,h_{Q-1}\bar{\}} \end{array}$$On the other hand,
(49)$$\begin{array}{ccc} \left({\bar{G}}_{d}\times h_{j^{*}}\right)\;\mathrm{mod}\;d\overset{\left(47\right)}{=}\left({\bar{G}}_{d}\times g_{i^{*}}\right)\;\mathrm{mod}\;d & = & \left(\left(G_{n}\;\mathrm{mod}\;d\right)\times g_{i^{*}}\right)\;\mathrm{mod}\;d \end{array}$$
(50)$$\begin{array}{ccc} & = & (G_{n}\times g_{i^{*}})\;\mathrm{mod}\;d \end{array}$$
(51)$$\begin{array}{ccc} & \overset{\left(43\right)}{=} & G_{n}\;\mathrm{mod}\;d \end{array}$$
(52)$$\begin{array}{ccc} & = & {\bar{G}}_{d}=\bar{\{}h_{1},\cdots ,h_{Q-1},\underset{|h_{Q}|\mathrm{times}}{\underset{︸}{h_{Q},\cdots ,h_{Q}}}\bar{\}} \end{array}$$

Comparing (Equation 48) and (Equation 52) (i.e., the number of times that $h_{Q}$ appears), we have proved that $|h_{1}\left|=\right|h_{Q}|$. The proof of the second part, and thus the proof of the lemma, are now complete.

## 4. Generalization

In this section, we consider several generalizations of the coding scheme presented in the previous section, to illustrate how the insights generalize beyond the basic setting.

### 4.1. Optimized Additive Randomness

In the coding scheme presented in Theorem 1, the additive common randomness *z* appeared in the codewords $X_{1},X_{2}$ is *uniform* over $F_{q}$ or $Z_{n}$ (refer to (Equation 7)), which is not necessary but a universal and convenient choice that works for all cases and admits a simple proof. We show, through the following example, that an optimized *z* (which does not have full-support over $Z_{n}$) might help to further reduce the communication cost.

**Example** **6.**
*Consider the function $f(W_{1},W_{2})$ shown in Figure 5, where a feasible expanded function over $Z_{4}$ is also depicted. The confusable sets are obtained from Theorem 3 using $G_{4}=Z_{4}^{\times }=\left\{1,3\right\}$.*

(53)
$$\begin{array}{c} Z_{4}^{\times }=\left\{0\right\}\cup \left\{1,3\right\}\cup \left\{2\right\},\;\gamma \;\mathrm{is}\;\mathrm{uniform}\;\mathrm{over}\;G_{4}=\left\{1,3\right\}. \end{array}$$

*From Theorem 1, Alice will send $X_{1}=\gamma \times {\tilde{W}}_{1}+z$ and Bob will send $X_{2}=\gamma \times {\tilde{W}}_{2}-z$ to Carol, where γ is uniform over {1,3}, z is uniform over $Z_{4}=\left\{0,1,2,3\right\}$, and γ,z are independent. That is, Alice and Bob each sends a symbols from $Z_{4}$ (i.e., 2 bits) to Carol.*

*Interestingly, if we choose z to be uniform over {0,2} (instead of uniform over {0,1,2,3}), the scheme will also work. Correctness remains the same and for security, we only need $\left(X_{1},X_{2}\right)$ (or equivalently $\left(X_{1},X_{1}+X_{2}\right)$) to be identically distributed when $\left(W_{1},W_{2}\right)\in \left\{\left(0,0\right),\left(0,1\right)\right\}$. Note that*

(54)
$$\begin{array}{ccc} & & \left(W_{1},W_{2}\right)=\left(0,0\right)\to \left({\tilde{W}}_{1},\tilde{W_{2}}\right)=\left(1,0\right)\to \left(X_{1},X_{1}+X_{2}\right)=\left(\gamma +z,\gamma \right),\\ & & \left(W_{1},W_{2}\right)=\left(0,1\right)\to \left({\tilde{W}}_{1},\tilde{W_{2}}\right)=\left(1,2\right)\to \left(X_{1},X_{1}+X_{2}\right)=\left(\gamma +z,\gamma \times 3\right). \end{array}$$

*Both (γ+z,γ) and (γ+z,γ×3) are uniform over {(1,1),(3,1),(3,3),(1,3)}. Therefore, the scheme satisfies the security constraint. Importantly, now $X_{2}=\gamma \times \tilde{W_{2}}-z$ can only take value 0 or 2. Therefore, Bob only needs to send 1 bit (instead of 2 bits) to Carol.*


**Remark** **3.**
*If variable-length codes are allowed, then the above code can be further improved. Specifically, Alice does not need to distinguish whether $X_{1}$ is 1 or 3, e.g., Alice may simply send 1 when $X_{1}$ is 1 or 3 (this happens when ${\tilde{W}}_{1}=1$). Interestingly, the rate region of this code coincides with an existing converse result from Theorem 9 in [6] for any joint distribution of $\left(W_{1},W_{2}\right)$ with full support. Thus, this improved code with optimized additive randomness and variable-length codewords tuns out to be information theoretically optimal (i.e., even if block codes are allowed). We also note that an alternative optimal code construction based on a different idea is presented in [6] (see Algorithm 3).*


For a general given function $f(W_{1},W_{2})$, to find the optimal choice of *z* of minimum randomness, we may list all identically distributed conditions in the security constraint (such as (Equation 54)) and solve for the *z* variable that satisfies all the constraints and has minimum entropy (a uniform full-support *z* will always work but has maximum entropy).

### 4.2. ϵ-Error Schemes with Block Codes

Hitherto, we have focused exclusively on scalar codes and zero-error schemes that work for any joint distribution of $\left(W_{1},W_{2}\right)$. In this subsection, we show how to use classical source coding techniques (specifically, structured linear codes, or Korner–Marton coding [34]) that exploit the specific distribution of $\left(W_{1},W_{2}\right)$ to improve the communication rate when long block codes and vanishing-error are allowed. This is explained through the following binary AND function example.

**Example** **7.**
*Consider the binary AND function $f\left(W_{1},W_{2}\right)=W_{1}\;AND\;W_{2}$, for which a feasible expanded function over $F_{3}$ is shown in Figure 6. The confusable sets are obtained from Theorem 2 using the primitive element g=2 of $F_{3}^{\times }$ and the divisor d=1.*

(55)
$$\begin{array}{c} F_{3}=\left\{0\right\}\cup \left\{2^{0},2^{1}\right\}=\left\{0\right\}\cup \left\{1,2\right\},\;\gamma \;\mathrm{is}\;\mathrm{uniform}\;\mathrm{over}\;\left\{1,2\right\}. \end{array}$$

*Then, from Theorem 1, we set $X_{1}=\gamma \times {\tilde{W}}_{1}+z,\;X_{2}=\gamma \times \tilde{W_{2}}-z$ so that it suffices to send a symbol from $F_{3}$ (i.e., $\log_{2}3$ bits) each from Alice and Bob to Carol. In other words, the rate tuple $\left(\log_{2}3,\log_{2}3\right)$ is achievable. As mentioned in the introduction, this zero-error scalar code first appeared in Appendix B of [12].*

*We note that for correct decoding, Carol will compute $X_{1}+X_{2}=\gamma \times \left(\tilde{W_{1}}+\tilde{W_{2}}\right)$, denoted by U. As our goal is only to recover U (securely of course), the amount of information required is simply the entropy of U (which is smaller than $\log_{2}3$ bits as long as it is not uniform). The only caveat is that encoding is done in a distributed manner at Alice and Bob respectively, so we just need to compress U with a linear code such that it is compatible with the decoding procedure of $X_{1}+X_{2}$. Fortunately, this distributed source compression for sum computation problem has been studied in network information theory. In particular, structured linear codes apply and we will use (the secure version of) Korner–Marton coding [34].*

*The improvement of the communication rate comes from the observation that in our proposed code, we consider the worst case, i.e., U is not compressed and a symbol from $F_{3}$ is sent to represent $X_{1}$ regardless of the distribution of U, the variable we wish to recover. When U is not uniform, further compression over long blocks is possible. As a simple example, suppose $W_{1}$ and $W_{2}$ are two independent uniform binary variables. As a result, $U=X_{1}+X_{2}=\gamma \times \left({\tilde{W}}_{1}+{\tilde{W}}_{2}\right)$ is 0 with probability 1/4, is 1 with probability 3/8, and is 2 with probability 3/8 (see Figure 6) and the entropy of U is*

$$\begin{array}{c} H\left(U\right)=H\left(\frac{1}{4},\frac{3}{8},\frac{3}{8}\right)=\frac{1}{4}\left(11-3\log_{2}3\right)<\log_{2}3. \end{array}$$

*Next, we outline how to use structured linear source codes to achieve the rate tuple $\left(R_{1},R_{2}\right)=\left(H\left(U\right)\log_{2}3,H\left(U\right)\log_{2}3\right)$ bits per input symbol over long block-length with vanishing probability of error. Consider L-length extension of the two inputs $W_{1},W_{2}$, denoted by $\vec{W_{1}},\vec{W_{2}}$, i.e., $\vec{W_{1}},\vec{W_{2}}$ are two sequences of i.i.d. uniform bits of length L. A similar vector notation is used for L-length extensions of other variables, e.g., $\vec{z}$ represents a length L sequence of i.i.d. uniform symbols over $F_{3}$. We apply our proposed scheme to each bit of the input sequence and then multiply (over $F_{3}$) the vector codeword with a matrix A of size (H(U)+ϵ)L×L.*

$$\begin{array}{ccc} & & {\vec{X}}_{1}=A_{(H(U)+\epsilon )L\times L}\cdot \left({\vec{\gamma }}_{L\times 1}\times \underset{L\times 1}{\underset{︸}{{\vec{\tilde{W}}}_{1}}}+{\vec{z}}_{L\times 1}\right),\;{\vec{X}}_{2}=A_{(H(U)+\epsilon )L\times L}\cdot \left(\vec{\gamma }\times {\vec{\tilde{W}}}_{2}-\vec{z}\right)\\ & \Rightarrow & {\vec{X}}_{1}+{\vec{X}}_{2}=A_{(H(U)+\epsilon )L\times L}\cdot (\underset{≜\vec{U}}{\underset{︸}{\vec{\gamma }\times \left({\vec{\tilde{W}}}_{1}+{\vec{\tilde{W}}}_{2}\right))}}=A_{(H(U)+\epsilon )L\times L}\cdot \vec{U} \end{array}$$

*where the ‘+’ and ‘×’ operators are symbol-wise, and the ‘·’ operator is the matrix multiplication operator. Further optimizations of the common randomness consumption are possible, i.e., the same randomizer can be used for each input bit and it suffices to use an additive common randomness variable with entropy $LH\left(U\right)\log_{2}3$ bits (instead of $L\log_{2}3$ bits). Note that the same matrix A must be used by both Alice and Bob. We need to ensure that from $A_{(H(U)+\epsilon )L\times L}\cdot \vec{U}$, we can recover $\vec{U}$. In other words, we now have the well-known point-to-point source coding problem with a linear compressor. Thus, there exists a deterministic matrix A of size (H(U)+ϵ)L×L such that we can recover $\vec{U}$ from $A\cdot \vec{U}$ with ϵ probability of error and ϵ→0 when L→∞. Specifically, a random generation of A (i.e., choosing each element of A independently and uniformly over $F_{3}$) will work with high probability. The structured linear coding technique has appeared in the literature many times, e.g., it was introduced by Elias in the context of channel coding over a binary symmetric channel [35], was used by Wyner in the context of distributed source coding of binary sources (the Slepian-Wolf problem, see Section VI. C of [36]), was used by Korner and Marton in the context of encoding module-two sum of binary sources [34], and generalizations to finite fields are immediate (see, e.g., in [37] and Remark 10.2 of [38]).*

*The security constraint is easily verified. The scalar code is secure by Theorem 1. Then independent application of the scalar code to L-length extensions is also secure. Multiplying with a deterministic matrix A will not leak any information.*

*Therefore, the optimized block code has vanishing probability of error and is secure. The achieved rate tuple is $\left(H\left(U\right)\log_{2}3,H\left(U\right)\log_{2}3\right)$ bits for the codewords per input bit.*

*Finally, we note that the idea of using Korner–Marton coding for secure computation is not new, e.g., it has been applied to secure sum computations [6,39]. While our objective is not sum computation, Korner–Marton coding still applies because the decoding procedure $X_{1}+X_{2}$ relies on a linear operation.*


**Remark** **4.**
*Interestingly, the zero-error code presented above for AND computation is information theoretically optimal in terms of the communication rate (refer to Theorem 11 of [6]). That is, communicating $\log_{2}3$ bits per input bit from Alice and Bob each to Carol is the minimum possible. Note that as ϵ-error codes achieve an improved rate performance than the best of that of zero-error codes, we know that for secure computation problems, ϵ-error capacity may be different from zero-error capacity (this fact has been established in prior work [6,39]). We also note that when ϵ-error is allowed, the optimal rate region for AND computation remains open.*


The above linear compression technique applies to all secure computation codes over $F_{q}$ (refer to Theorem 2), i.e., instead of sending a symbol from $F_{q}$, we may compress it to $H\left(U\right)\log_{2}q$ bits. However, we note that the same result does not hold for codes over $Z_{n}$ when *n* is not a prime. While the same linear compression technique can be applied, the rate performance is not known. That is, it is not known how large the matrix *A* needs to be, if we wish to recover $\vec{U}$ from $A\cdot \vec{U}$. In particular, H(U) may not suffice (we need to understand more on the open problem of source coding with restricted encoding structures, e.g., modular arithmetic. For related results on source coding with group codes, see, e.g., in [40,41] and references therein).

### 4.3. Equality Function with Non-Prime-Power Inputs

In this subsection, we continue the discussion on the equality function in the introduction. We consider the equation function with arbitrary input size, i.e., $W_{1},W_{2}\in \left\{0,1,\cdots ,m-1\right\}$ for an arbitrary integer *m* and wish to securely compute if $W_{1}$ is equal to $W_{2}$. The approach taken in the introduction works when *m* is a prime power $p^{n}$ such that there exist an invertible mapping between $\left\{0,1,\cdots ,p^{n}-1\right\}$ and the elements from the finite field $F_{p^{n}}$ (say $W_{1}\left(W_{2}\right)$ is mapped to $\tilde{W_{1}}\left(\tilde{W_{2}}\right)$). Then, $\tilde{W_{1}}-\tilde{W_{2}}$ is a feasible expanded function where the zero element and the non-zero elements are two confusable sets. Now, what if *m* is not a prime power? We may increase *m* to a prime power and then use the previous approach. This approach will require that Alice and Bob each sends a symbol of size larger than $\log_{2}m$ bits. Interestingly, we show that $\log_{2}m$ bits are always sufficient for any *m* (no matter whether *m* is a prime power or not). To this end, we need to use a variant of the expand-and-randomize scheme from Theorem 1. To illustrate the idea, in the following we consider the simplest example where *m* is not a prime power, i.e., m=6.

**Example** **8.**
*Consider $W_{1},W_{2}\in \left\{0,1,\cdots ,5\right\}$. $f(W_{1},W_{2})=\mathrm{Yes}$ if $W_{1}$ is equal to $W_{2}$ and otherwise $f(W_{1},W_{2})=\mathrm{No}$. While 6 is not a prime power, we may decompose it into products of prime powers, i.e., 6=2×3. The following scheme works by using a product of our expand-and-randomize schemes over decomposed domains with uniformly permuted inputs, i.e., using two schemes of equality function with prime inputs in parallel.*

*Alice and Bob share a common random variable $Z=(\pi ,\gamma_{1},\gamma_{2},z_{1},z_{2})$, where all the random variables are independent and uniform, π is from the set of all possible permutations with 6 elements, $\gamma_{1}$ is from $F_{2}^{\times }=\left\{1\right\}$, $z_{1}$ is from $F_{2}=\left\{0,1\right\}$, $\gamma_{2}$ is from $F_{3}^{\times }=\left\{1,2\right\}$, and $z_{2}$ is from $F_{3}=\left\{0,1,2\right\}$. The codewords are*

$$\begin{array}{c} X_{1}=\left(a_{1},a_{2}\right),\;where\;a_{1}\overset{F_{2}}{=}\gamma_{1}\times \left(\pi \left(W_{1}\right)\;\mathrm{mod}\;2\right)+z_{1},a_{2}\overset{F_{3}}{=}\gamma_{2}\times \left(\pi \left(W_{1}\right)\;\mathrm{mod}\;3\right)+z_{2};\\ X_{2}=\left(b_{1},b_{2}\right),\;where\;b_{1}\overset{F_{2}}{=}\gamma_{1}\times \left(\pi \left(W_{2}\right)\;\mathrm{mod}\;2\right)+z_{1},b_{2}\overset{F_{3}}{=}\gamma_{2}\times \left(\pi \left(W_{2}\right)\;\mathrm{mod}\;3\right)+z_{2}. \end{array}$$

*The ‘+’ and ‘×’ operations in computing $a_{1},b_{1}\left(a_{2},b_{2}\right)$ are over $F_{2}\left(F_{3}\right)$. Note that the same permutation π is applied to $W_{1}$ and $W_{2}$. The decoding rule of Carol is as follows.*

(56)
$$\begin{array}{c} \;Carol\;claims\;W_{1}is\;equal\;to\;W_{2}\;if\;and\;only\;if\;\left(a_{1},a_{2}\right)=\left(b_{1},b_{2}\right). \end{array}$$

*We have zero error because $\pi \left(W_{1}\right)=\pi \left(W_{2}\right)$ if and only if $\left(\pi \left(W_{1}\right)\;\mathrm{mod}\;2,\pi \left(W_{1}\right)\;\mathrm{mod}\;3\right)=\left(\pi \left(W_{2}\right)\;\mathrm{mod}\;2,\pi \left(W_{2}\right)\;\mathrm{mod}\;3\right)$ (this result is typically referred to as the Chinese Remainder Theorem). Next, we verify that the security constraint is satisfied, i.e., when $W_{1}$ is not equal to $W_{2}$, $X_{1},X_{2}$ are identically distributed. Consider any $\left(W_{1},W_{2}\right)$ such that $W_{1}\neq W_{2}$ and $\pi \left(W_{1}\right)\neq \pi \left(W_{2}\right)$. Note that $\left(X_{1},X_{2}\right)$ is invertible to $\left(a_{1},a_{2},b_{1}-a_{1},b_{2}-a_{2}\right)$, and the 3 variables $a_{1},a_{2},\left(b_{1}-a_{1},b_{2}-a_{2}\right)$ are independent. Further, $a_{1}$ is uniform over {0,1}, $a_{2}$ is uniform over {0,1,2}, and $\left(\pi \left(W_{1}\right),\pi \left(W_{2}\right)\right)$ is uniform over (i,j),i≠j,i,j∈{0,1,2,3,4,5} so that*

(57)
$$\begin{array}{ccc} & & \mathrm{Pr}\left(\pi \left(W_{1}\right)\;\mathrm{mod}\;2=\pi \left(W_{2}\right)\;\mathrm{mod}\;2,\pi \left(W_{1}\right)\;\mathrm{mod}\;3\neq \pi \left(W_{2}\right)\;\mathrm{mod}\;3\right)=2/5, \end{array}$$


(58)
$$\begin{array}{ccc} & & \mathrm{Pr}\left(\pi \left(W_{1}\right)\;\mathrm{mod}\;2\neq \pi \left(W_{2}\right)\;\mathrm{mod}\;2,\pi \left(W_{1}\right)\;\mathrm{mod}\;3=\pi \left(W_{2}\right)\;\mathrm{mod}\;3\right)=1/5, \end{array}$$


(59)
$$\begin{array}{ccc} & & \mathrm{Pr}\left(\pi \left(W_{1}\right)\;\mathrm{mod}\;2\neq \pi \left(W_{2}\right)\;\mathrm{mod}\;2,\pi \left(W_{1}\right)\;\mathrm{mod}\;3\neq \pi \left(W_{2}\right)\;\mathrm{mod}\;3\right)=2/5 \end{array}$$


(60)
$$\begin{array}{ccc} & & \Rightarrow \left(b_{1}-a_{1},b_{2}-a_{2}\right)\;\mathrm{is}\;\mathrm{uniform}\;\mathrm{over}\;\left\{\left(0,1\right),\left(0,2\right),\left(1,0\right),\left(1,1\right),\left(1,2\right)\right\}. \end{array}$$

*Therefore, security is guaranteed and sending $1+\log_{2}3=\log_{2}6$ bits from Alice and Bob to Carol each is sufficient.*


**Remark** **5.**
*The above scheme generalizes in a straightforward manner to any integer m using a prime-power decomposition of m (this result is typically referred to as the fundamental theorem of arithmetic). As the achieved communication rate $\log_{2}m$ bits for Alice and Bob each matches the optimal rate for independent and uniform inputs with no security constraint (see Proposition 1), the above secure computation scheme achieves the information theoretically optimal rate region.*


## 5. Discussion

We introduced the expand-and-randomize scheme for the secure computation problem and implement it over the finite field and the ring of integers modulo *n*. We characterized the algebraic structures of the feasible expanded functions through the notion of confusable sets. We find it interesting that while we consider only information theoretic security, the tools invoked from algebra and number theory arise frequently and lie in the core of cryptography under computational security (see textbooks, e.g., in [42,43]). The proposed scheme is very efficient and sometimes optimal when the original function is (close to) an isomorphism of such confusable sets. In particular, the information theoretically optimal communication cost is characterized for a new class of functions that include an infinite number of instances (refer to Proposition 1). However, we are also aware of functions where there exist better schemes such that our scheme is strictly sub-optimal. In the following, we present two such examples to expose more diverse insights for the challenging open problem of minimal secure computation.

### Sub-Optimal Examples

**Example** **9.**
*Consider the function shown in Figure 7. A feasible expanded function over $F_{7}$ is also depicted. The confusable sets are obtained from Theorem 2 using the primitive element g=3 of $F_{7}^{\times }$ and the divisor d=2.*

$$\begin{array}{c} F_{7}=\left\{0\right\}\cup \left\{3^{0},3^{2},3^{4}\right\}\cup \left\{3^{1},3^{3},3^{5}\right\}=\left\{0\right\}\cup \left\{1,2,4\right\}\cup \left\{3,5,6\right\},\;\gamma \;\mathrm{is}\;\mathrm{uniform}\;\mathrm{over}\;\left\{1,2,4\right\}. \end{array}$$

*Therefore, according to Theorem 1, it suffices to send a symbol from $F_{7}$ (i.e., $\log_{2}\left(7\right)$ bits) each from Alice and Bob to Carol. In other words, the rate tuple $\left(\log_{2}\left(7\right),\log_{2}\left(7\right)\right)$ is achievable.*

*However, an improved rate tuple (2,2) is achievable using a different coding scheme. Specifically, we use the coding scheme from Section 2 of [12]. The scheme (when applied to this example) is described as follows. The common random variable shared is $Z=(\pi ,z_{1},z_{2})$, where $\pi ,z_{1},z_{2}$ are independent and uniform binary random variables. The codewords are*

$$\begin{array}{c} X_{1}=\left\{\begin{array}{cc} \left(1,z_{1}\right) & when\;\pi =0,W_{1}=0\\ \left(2,z_{2}\right) & when\;\pi =0,W_{1}=1\\ \left(2,z_{1}\right) & when\;\pi =1,W_{1}=0\\ \left(1,z_{2}\right) & when\;\pi =1,W_{1}=1 \end{array}\right.,\;X_{2}=\left\{\begin{array}{cc} \left(f\left(0,W_{2}\right)+z_{1},f\left(1,W_{2}\right)+z_{2}\right) & when\;\pi =0\\ \left(f\left(1,W_{2}\right)+z_{2},f\left(0,W_{2}\right)+z_{1}\right) & when\;\pi =1 \end{array}\right. \end{array}$$

*where $X_{1}$ and $X_{2}$ each contains 2 bits. The coding idea is that the first (second) row of the function table is protected by the uniform noise $z_{1}$ ($z_{2}$). Bob does not know the value of $W_{1}$, so he will send both $f(0,W_{2})$ and $f(1,W_{2})$ (after masked by $z_{i}$) in a random order. Alice will use the first element of $X_{1}$ to indicate if the first or the second element of $X_{2}$ contains the desired function and use the second element to carry the noise that masks the desired function output. Therefore, the decoding rule is as follows. Denote $X_{1}=\left(a_{1},a_{2}\right)$ and $X_{2}=\left(b_{1},b_{2}\right)$. Carol can recover $f(W_{1},W_{2})$ with no error from $b_{a_{1}}-a_{2}$. Security is guaranteed by the observation that Carol only knows the noise that masks the desired function while obtains nothing else. Therefore in this scheme, sending $2<\log_{2}7$ bits each by Alice and Bob is sufficient.*


**Example** **10.**
*Consider the function shown in Figure 8. A feasible expanded function over $Z_{8}$ is also depicted. The confusable sets are obtained from Theorem 3 using the subgroup $G_{8}=\left\{1,3\right\}$ of $Z_{8}^{\times }=\left\{1,3,5,7\right\}$.*

(61)
$$\begin{array}{c} Z_{8}=\left\{0\right\}\cup \left\{1,3\right\}\cup \left\{5,7\right\}\cup \left\{2,6\right\}\cup \left\{4\right\},\;\gamma \;\mathrm{is}\;\mathrm{uniform}\;\mathrm{over}\;G_{8}=\left\{1,3\right\}. \end{array}$$

*Therefore, according to Theorem 1, it suffices to send a symbol from $Z_{8}$ (i.e., 3 bits) each from Alice and Bob to Carol. In other words, the rate tuple (3,3) is achievable.*

*However, an improved rate tuple is achievable using a different coding scheme. To see this, we assume that $W_{1},W_{2}$ are independent and each of them is uniform over its support (note that the scheme above does not depend on the joint distribution of $\left(W_{1},W_{2}\right)$). We now describe a scheme that achieves the rate tuple $\left(2,\log_{2}3\right)$, which is strictly better than (3,3). This scheme is inspired by Algorithm 3 of [6], which is for the function in Example 6 and we generalize it to the function in Figure 8. Alice and Bob share the common random variable $Z=(z,z^{\prime })$, where $z,z^{\prime }$ are independent uniform binary random variables. The codewords sent are*

(62)
$$\begin{array}{c} X_{1}=\left\{\begin{array}{cc} W_{1},z^{\prime } & \;when\;W_{1}=0\\ W_{1},z & \;when\;W_{1}=1 \end{array}\right.,\;X_{2}=\left\{\begin{array}{cc} {\left(W_{2}+z\right)}_{F_{2}} & \;when\;W_{2}\in \left\{0,1\right\}\\ 2 & \;when\;W_{2}=2 \end{array}\right. \end{array}$$

*where $X_{1}$ contains 2 bits and $X_{2}$ contains $\log_{2}3$ bits. Obviously $z^{\prime }$ is useless and it appears here to produce a fixed-length code. An important feature of this function is that from $f(W_{1},W_{2})$, we can always recover $W_{1}$. Therefore, $W_{1}$ is always sent from Alice. Further, $W_{2}$ should be protected when $W_{2}\in \left\{0,1\right\}$ and $W_{1}=0$. Thus, $W_{2}$ is protected by a uniform noise z and when it should be revealed (i.e., when $W_{1}=1$), Alice will send the noise z; otherwise, when it should be protected, Alice will send an independent (thus useless) noise $z^{\prime }$. After the coding idea is explained, the decoding rule is now obvious. Carol will first check the value of $W_{1}$ (sent from Alice). If $W_{1}=0$, Carol will claim f=0 if $X_{2}\in \left\{0,1\right\}$ and f=1 if $X_{2}=2$. If $W_{1}=1$, Carol will claim f=4 if $X_{2}=2$ and otherwise if $X_{2}\in \left\{0,1\right\}$, Carol will use z to recover $W_{2}$ and based on $\left(W_{1},W_{2}\right)$, f is decoded with no error. Security is guaranteed because when $\left(W_{1},W_{2}\right)\in \left\{\left(0,0\right),\left(0,1\right)\right\}$, for both cases we have that $X_{1},X_{2}$ are independent, in $X_{1}=\left(W_{1},z^{\prime }\right)$, $W_{1}$ is fixed to 0, $z^{\prime }$ is uniform and $X_{2}=W_{2}+z$ is also uniform. Therefore, using this improved scheme, it suffices to send 2<3 bits by Alice and $\log_{2}3<3$ bits by Bob, respectively.*


Going forward, while the characterization of the algebraic structures of the confusable sets over $F_{q}$ and $Z_{n}$ is given, i.e., the confusable sets in Theorems 2 and 3 are complete (a simple consequence of the confusable sets definition is the closure property under multiplication), the algorithmic aspect of the expand-and-randomize scheme over $F_{q}$ and $Z_{n}$ is wide open, i.e., we do not have efficient algorithms that can help us quickly identify (minimal) feasible expanded functions when they exist. The solutions to current examples are mainly found through the lists of confusable sets in Appendix A and Appendix B. Efficient algorithms that can classify functions and realize associations with isomorphisms of confusable sets are of immediate interest, but appear challenging at the same time. Going beyond the finite field and the ring of integers modulo *n*, it is interesting to explore other widely studied algebraic objects in abstract algebra [23], e.g., the matrix ring and the polynomial ring. Generally speaking, the expand-and-randomize scheme captures the idea of *embedding* the function to compute in another function that guarantees security. The potential of this general embedding theme remains to be fully explored. Finally, we note that while we focus on the basic model of minimal (non-interactive three-user) secure computation, the proposed scheme generalizes immediately to interactive protocols (by first interactively generating the common randomness) and to more users (the notions of expanded functions generalize in a natural manner). Exploration of the proposed scheme to various models of secure computation [11] is an interesting research avenue.

## Figures and Tables

**Figure 1 entropy-23-01461-f001:**
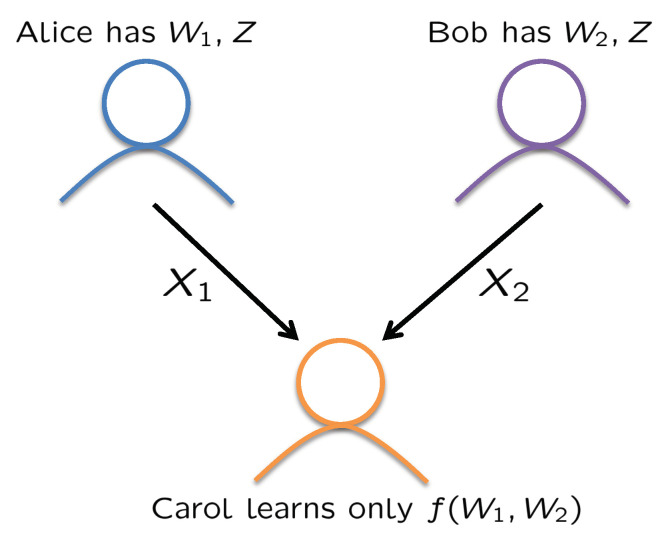
The minimal secure computation problem [12].

**Figure 2 entropy-23-01461-f002:**
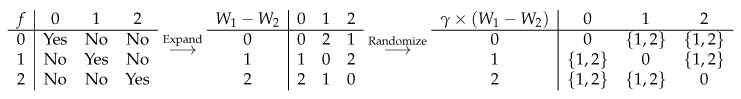
The expand-and-randomize coding scheme for the *equal* function. In the function table, each row corresponds to a value of $W_{1}$ and each column corresponds to a value of $W_{2}$. When the function output is a random variable, the set of possible values is shown in the table. The functions $W_{1}-W_{2}$ and $\gamma \times (W_{1}-W_{2})$ are computed over the finite field $F_{3}$ and γ is uniform over the set {1,2}.

**Figure 3 entropy-23-01461-f003:**
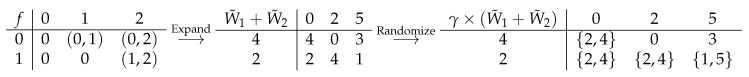
The expand-and-randomize coding scheme for the *selected-switch* function. The expanded function ${\tilde{W}}_{1}+{\tilde{W}}_{2}$ and the randomized expanded function $\gamma \times ({\tilde{W}}_{1}-{\tilde{W}}_{2})$ are defined over the ring of integers modulo 6, $Z_{6}$. γ is uniform over {1,5}, the set of integers that are coprime with 6.

**Figure 4 entropy-23-01461-f004:**
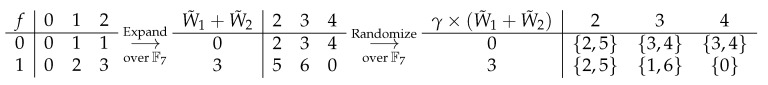
An expand-and-randomize secure computation code over $F_{7}$, where γ is uniform over {1,6}.

**Figure 5 entropy-23-01461-f005:**
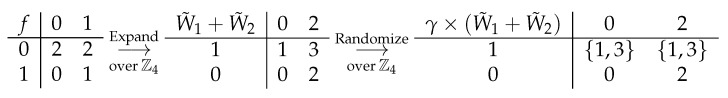
An expand-and-randomize secure computation code over $Z_{4}$, where γ is uniform over {1,3}.

**Figure 6 entropy-23-01461-f006:**
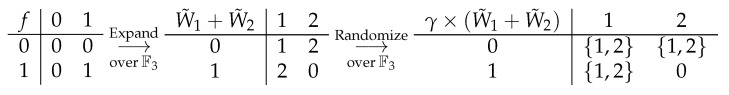
An expand-and-randomize secure computation code (for the binary AND function) over $F_{3}$, where γ is uniform over {1,2}.

**Figure 7 entropy-23-01461-f007:**
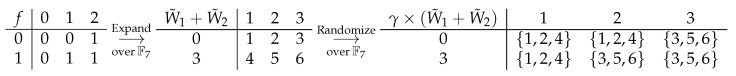
An expand-and-randomize secure computation code over $F_{7}$, where γ is uniform over {1,2,4}.

**Figure 8 entropy-23-01461-f008:**
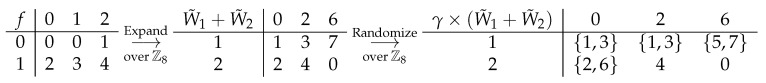
An expand-and-randomize secure computation code over $Z_{8}$, where γ is uniform over {1,3}.

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
