# Peer review of "Expand-and-Randomize: An Algebraic Approach to Secure Computation"

_entropy, 2021, doi:10.3390/e23111461_

Round 1

Reviewer 1 Report

The topic of the paper provides new protocol design ideas for a special case of non-interactive multiparty computation, called Private Simultaneous Messages (PSM), where the considered security is information theoretical. Such protocols exist for every function over finite domain, see [12,13], but they need exponentially large messages. The paper discusses methods for designing efficient PSM protocols and illustrates it with numerous small examples. The general idea is these protocols is the following. The message space is a (multiplicative) group with operation * on which another commutative group operation + is defined. Alice and Bob has private inputs W1 and W2, respectively, and get γ and z as private random values. Alice's message is (γ*W1+z), while Bob' message is (γ*W2-z). Charlie computes the sum of the received values, which he announces as the result. Here "z" is used to hide the values sent by the parties, while γ is to ensure privacy after the function's value is computed. This protocol works under quite general assumptions as stated in Theorem 1. The main bulk of the paper discusses two instantiations. First, multiplication and addition are operations on a finite field; second, they are operations modulo a (composite) integer. For both cases examples of "confusable set system" are given with proofs that they are indeed such systems. The appendix enlists the first few such confusable set systems. Possible generalizations and side issues are discussed in Section 4, and the conclusion is in Section 5.

All results are correct, the topic is current and interesting. This reviewer, however, thinks that the presentation is too lengthy, and unnecessarily repetitive. Minor claims are proved, almost every notion is supplemented by many examples. Important remarks vanish in the lengthy exposition. I think sometimes less is more.

Apart from these general remarks, a long list of minor remarks follow. There are several formulas outside the margins, and even more formulas overlapping formula numbers. These should be formatted correctly. The paper uses italics for remarks and examples. Italics is used for emphasis only typically for a single sentence (such as a statement of a theorem), and not for long exploration. Please don't abuse italics. It is hard to read.

Minor remarks:

Footnote 1: Never start a footnote with ellipsis.
Line 41: "...while efficient codes are mostly not available"  Please rewrite
Line 60: "not supposed to learn *whether* W1-W2 is"
Line 72:  "z" appears suddenly, however its presence is essential.
Lines 72-92: Too long discussion, what is explained here is clear.
Line 92: "Remarkably" ???
Line 106: "step is fulfilled" ???
Line 121: formula overlaps the formula number
Lines 122-137: shorten significantly

The Introduction actually explains everything which is covered in Sections 2 and 3, thus mainly it is a repetition of later material. While some repetition is good, this one seems to be too extensive.

Line 138: Footnote 2 should go after Section head into the main text.
Line 145: introducing L1 and L2 is problematic, as the paper uses "z" to hide the private inputs from Alice and Bob, thus L1 and L2 should be equal.
Line 157: reword the sentence to avoid "n,Z_n"
Line 161: This definition is information-theoretic, or computational? In the latter case S^* should be effectively sampleable. 
Line 162: It is easy to miss that the function f is fixed to be the "addition".
Line 169: "equal function" => "equality function"
Footnote 4: Over a finite field every function is polynomial. Please rewrite
Line 178: Theorem 1 is clear, no need for such a long explanation. Please denote END-OF-PROOF, not only here, but everywhere. Add remarks about the necessity of "z" which is ignored.
Line 188: \gamma is missing before X1+X2
Section 3.1: Readers can be assumed familiar with finite fields, generator polynomials, and addition / multiplication on finite fields.
Line 218: Compress the statement of Theorem 2.
Line 220-231: No need for this Example and Remark. Compress into a single paragraph if you think it is absolutely necessary. Don't use italics for Example and Remark.
Line 220: Don't use italics; reserve italics for statements. Finding a primitive element can be easy (pick an element randomly and check). Reword "extremely heavy" (maybe "difficult"?)
Line 231: Mark the end of the proof. Compress or omit the proof, it is trivial form the definitions.

Subsection 3.2: Properties of the multiplicative group Z^x_n can be assumed to be known. Try to compress this description. When defining sets, use the shorter \{a \in Z_n | gcd(a,n)=d\} format.
Line 256: No need to prove this lemma, this is well-known (elementary algebra). Using the notation G_n and G_d is misleading, try to figure out another notation.
Line 262: don't use italics
Line 265: delete "(note that ... )"
Line 272: The notation G_n, G_d, G_{n/d} are confusing, while the meaning is clear.
Line 276: No italics
Line 286: The notation G_n and G_d, as well as d_1,..d_b are unfortunate.
Lines 288-308: Shorten. Theorem 3 is clear.
Line 290: No italics. Formula overlaps the formula number.
Line 308: mark the end of the proof
Line 309: No italics.

Subsection 3.3: The converse

"Converse" statements are in coding theory. In cryptography such statements are considered only in asymptotic sense (if at all), and, apart from some interesting facts, this section gives not too much. The additional privacy constraints require additional masking, which might require balancing of the two sources. While the discussion after Proposition 1 is too long, it would help to add a sentence on what converse is.

Section 3.4: The reviewer suggest omitting this section completely .

Section 4: This section contains very interesting and relevant material. Examples, remarks should not be in italics (consider not calling them examples or remarks). Long explanations should be truncated (e.g, lines 405-408 can be omitted). 
Line 410: "..does not need to distinguish *whether*"
Line 417: some words about why "z" is here.
Line 449: why don't use 1/4(11-3\log_2 3)  and not in 3-ary units?
Lines 450-469: Shorten
Lines 470-473: Not needed.
Line 510 and on: No italics, please. Here the authors use the simple idea of doing two instances of the protocol in parallel. Maybe it is worth to mention separately. The explanation then could be shortened.

Section 5: Conclusion

I suggest using past tense for this paragraph. Don't use italics for the examples.

Reviewer 2 Report

The manuscript is interesting and well written. Another merit consists, in my opinion, in the readability and simplicity of the arguments, all for the benefit of the reader. After a minor revision, see for example some points below, I could recommend the paper for publication on Entropy.

(1) Page 1, line 33: please, rephrase the sentence better. In particular "are independent". (2) Page 4: compress Eq. 5 a little, or break it into two lines.The same for Eq. (7) page 5, Eq. (49) page 12 and others. (3) Page 4, lines 133 and 134: "Note that... a prime p." Can you rewrite this sentence better? (4) Page 4, line 140: recall what is |f|. (5) Page 5, line 156: the symbol F_q seems to have not been explained before. (6) Page 7: why don't you try to write the statement of Theorem 2 in a more elegant and readable way? (7) Some equation numbers are not necessary: they are not recalled in the text (see, for example (6) and (55)). Check them all and remove unnecessary ones.
